# Time-series cardiovascular risk factors and receipt of screening for breast, cervical, and colon cancer: The Guideline Advantage

Aixia Guo[1], Bettina F. Drake[2], Yosef M. Khan[3], James R. Langabeer II[4], Randi E. Foraker[1,5]*

1 Institute for Informatics (I2), Washington University School of Medicine, St. Louis, MO, United States of America, 2 Division of Public Health Sciences, Department of Surgery, Washington University in St. Louis School of Medicine, St. Louis, MO, United States of America, 3 Health Informatics and Analytics, Centers for Health Metrics and Evaluation, American Heart Association, Dallas, TX, United States of America, 4 School of Biomedical Informatics, Health Science Center at Houston, The University of Texas, Houston, TX, United States of America, 5 Department of Internal Medicine, Washington University School of Medicine, St. Louis, MO, United States of America

* randi.foraker@wustl.edu

## Abstract

### Background

Cancer is the second leading cause of death in the United States. Cancer screenings can detect precancerous cells and allow for earlier diagnosis and treatment. Our purpose was to better understand risk factors for cancer screenings and assess the effect of cancer screenings on changes of Cardiovascular health (CVH) measures before and after cancer screenings among patients.

### Methods

We used The Guideline Advantage (TGA)—American Heart Association ambulatory quality clinical data registry of electronic health record data (n = 362,533 patients) to investigate associations between time-series CVH measures and receipt of breast, cervical, and colon cancer screenings. Long short-term memory (LSTM) neural networks was employed to predict receipt of cancer screenings. We also compared the distributions of CVH factors between patients who received cancer screenings and those who did not. Finally, we examined and quantified changes in CVH measures among the screened and non-screened groups.

### Results

Model performance was evaluated by the area under the receiver operator curve (AUROC): the average AUROC of 10 curves was 0.63 for breast, 0.70 for cervical, and 0.61 for colon cancer screening. Distribution comparison found that screened patients had a higher prevalence of poor CVH categories. CVH submetrics were improved for patients after cancer screenings.

**Data Availability Statement:** The data are owned by a third party and the authors do not have permission to share the data. Requesting access to The Guideline Advantage (TGA) data must be done

by contacting the American Heart Association via email: qualityresearch@heart.org. The Python code related to the analyses can be found in Github repository: https://github.com/aixiaguo/Python-Code.

**Funding:** The author(s) received no specific funding for this work.

## Conclusion

Deep learning algorithm could be used to investigate the associations between time-series CVH measures and cancer screenings in an ambulatory population. Patients with more adverse CVH profiles tend to be screened for cancers, and cancer screening may also prompt favorable changes in CVH. Cancer screenings may increase patient CVH health, thus potentially decreasing burden of disease and costs for the health system (e.g., cardio-vascular diseases and cancers).

## Introduction

Cancer is the second leading cause of death for both men and women in the United States (US) [1]: breast cancer is the second leading cause of cancer death among women [2]; colorectal cancer ranks second among men and third among women [3]; while cervical cancer ranks as a major cause of cancer death among women [4]. Regular cancer screenings for breast, cervical, and colorectal cancers can help to diagnose cancers early and reduce cancer deaths [5]. For example, in the past 40 years, the number of deaths caused by cervical cancer has significantly decreased thanks to pap tests which can find abnormal cervical cells before they turn to cancer [6]. Similarly, colonoscopy removes non-cancerous colon polyps before becoming malignant. And regular mammography screening can identify breast cancer in an earlier, more treatable stage. Thus, breast cancer screening (BCS), cervical cancer screening (CECS), and colorectal cancer screening (COCS) are very important for early detection and treatment.

Factors associated with cancer screenings include: demographic factors, health insurance coverage, education level, smoking status, obesity, and cholesterol testing. For example, receipt of mammography is associated with modifiable factors such as weight, smoking, and other lifestyle factors [7–11]. Receipt of CECS is associated with healthier weight [12], lower cardiovascular disease occurrence [13], and lower cholesterol [14]. Some studies suggest that smoking, sedentary lifestyle, high body mass index, and high comorbidity are associated with a higher percentage of COCS participation [15–17]. Traditionally, data for such studies originate from questionnaires, claims data, and telephone surveys, and statistical analysis methods such as logistic regression models are applied to examine the associations between the risk factors and cancer screenings. Electronic health records (EHR) contain longitudinal healthcare information and data including diagnoses, medications, procedures, lab tests, and images [18] and therefore can be used to discover new patterns and relationships from the rich data. Deep learning algorithms have been widely and successfully used in bioinformatics and healthcare fields as they can effectively capture features and patterns in longitudinal data [19,20].

In this study, we investigated associations between longitudinal CVH risk factors and the receipt of cancer screenings using EHR data by the long short-term memory (LSTM) model [21]. We then studied the distribution of CVH factors between patients who did and did not receive cancer screenings to further investigate the associations. Finally, we compared measures of CVH longitudinally within those who did and did not receive screening to better understand the effect of cancer screenings on CVH measures.

## Materials and methods

### Ethics statement

All the data were fully anonymized before we accessed them. Our study was approved by the Institutional Review Board at the Washington University School of Medicine in St. Louis. We

obtained a written acknowledgement of proprietary rights and non-disclosure and data use agreement from the American Heart Association (The Washington University_NDA_DUA_-CONTRACTID 158065_2019.04.26_K).

## Data source and study population

The Guideline Advantage (TGA) is a clinical data registry established in 2011 by the American Cancer Society, the American Diabetes Association, and the American Heart Association (AHA) [22]. EHR data has been collected from over 70 clinics across the US by the TGA to track and monitor disease management and outpatient preventative care [23]. We used longitudinal TGA data to predict three types of cancer screenings among 362,533 unique patients.

We used a 6-year range (2010–2015) to identify 777 female patients in the 40–69 year old age group who received BCS; 617 female patients in the 21–64 year old age group who received CECS; and 264 patients in the 50–75 year old age group who received COCS. If patients received multiple types of cancer screening, we only considered the first. Using the same criteria for gender and age, we randomly selected a comparison group of patients who did not receive cancer screenings: 8000 for BCS, 6000 for CECS, and 3000 for COCS.

We utilized the following CVH measures defined by the AHA: smoking status, body mass index (BMI), blood pressure (BP), hemoglobin A1c (A1C), and cholesterol (Low-Density Lipoprotein (LDL) in our dataset). We then classified them into three categories: ideal, intermediate, or poor, according to Table 1. We utilized the Multum drug database [24] as a template to convert the drug names in our dataset to their corresponding drug classes. The Levenshtein distance algorithm [25] was employed for the conversion by comparing the drug names in our dataset to the Multum drug database template. The conversion was considered successful and medications were considered as treatments for BP, A1C, or LDL (Table 1) if the distance between the two compared strings was less than five. All CVH measurements prior to the date of cancer screening were considered in the analysis for those who received screening, and all CVH measurements in the data set were considered in the analysis for those who did not receive screening.

For the primary analysis, we selected patients who had at least one measure of CVH: 725 for BCS, 565 for CECS, and 240 for COCS. In the comparison groups, there were available data for 8,000 BCS; 3,548 CECS; and 3,000 COCS.

## Statistical analysis

We first studied the LSTM prediction of cancer screening from time-series CVH factors. We divided each CVH factor into its submetric of "ideal", "intermediate", or "poor" according to Table 1. For example, if a patient had a measure of "ideal" blood pressure, then that feature

**Table 1. Measures of CVH which are available in the TGA (Adapted from: Lloyd-Jones, 2011) [26].**

|  | Poor Health | Intermediate Health | Ideal Health |
|---|---|---|---|
| Health Behaviors |  |  |  |
| Smoking status | Yes | Former ≤ 12 months | Never or quit > 12 months |
| Body mass index | ≥ 30 kg/m$^2$ | 25–29.9 kg/m$^2$ | < 25 kg/m$^2$ |
| Health Factors |  |  |  |
| LDL | ≥ 160 mg/dL | 130–159 mg/dL or treated to goal | < 130 mg/dL |
| Blood pressure | Systolic ≥ 140 mm Hg or Diastolic ≥ 90 mm Hg | Systolic 120–139 mm Hg or Diastolic 80–89 mm Hg or treated to goal | Systolic < 120 mm Hg Diastolic < 80 mm Hg |
| Fasting plasma glucose | ≥ 126 mg/dL | 100–125 mg/dL or treated to goal | < 100 mg/dL |

was called blood pressure ideal. All features were then embedded to a 32-dimensional vector space by word2vec [27] for each type of cancer screenings. The Python Genism Word2Vec model used the following hyperparameters: size (embedding dimension) was 32, window (the maximum distance between a target word and all words around it) was 5, min_count (the minimum number of words counted when training the model) was 1, sg (the training algorithm) was CBOW (the continuous bag of words). Time information for each measure was added and was calculated by the difference in days between each visit date and the most recent visit date. Thus, each feature was associated with its own time point in the unit of days.

The resulting embedded vectors and associated time points were fed to the LSTM model. Due to the comparison group being much larger than the number of patients with cancer screening, we randomly selected 800 patients for BCS, 600 patients for CECS, and 300 patients for COCS and repeated this process for 10 times to account for the imbalance between screened and unscreened groups. Each time, the data set for each type of cancer screening was split into a training data set (80%) and a test data set (20%). We trained the LSTM model on the training data and tested the trained model on the test data. We utilized the average of the area under the receiver operator curve (AUROC) to evaluate the performance of our LSTM model for each type of cancer evaluated.

Our LSTM model comprised an input layer, one hidden layer (with 100 dimensions) and an output layer. The hyperparameter used in the model was as follows: a sigmoid function was used as the activation function in the output layer. A binary cross-entropy was used as the loss function. Adam optimizer [28] was used to optimize the model with a mini-batch size of 64 samples.

We then investigated whether distributions of CVH–counts and percentages for each submetric–differed between patients who did and who did not receive cancer screenings by Chi-Squared test. Finally, we studied changes in CVH factors within screening group, for the same patients who received screening and for those who did not. Within screening group, we compared CVH measures from before and on the day of the screening to the CVH measures collected after the screening. For the patients who did not receive screening, we compared CVH measures before and after the mid-point of the visit dates. If patients only had a single visit, then they were not included in the before and after analysis. Analyses were conducted by using the libraries of Scikit-learn, Scipy, Matplotlib with Python, version 3.6.5 in 2019.

## Results

The majority of our study population was white, with a mean of age of approximately 55 years for BCS, 50 years for CECS, and 60 years for COCS (Table 2). The non-white study population was predominantly African-American. The average number of measures (*Avg #*) among patients who received screening was higher than that of patients who were not screened. For example, the average number of BP measurements for patients with BCS was 11 (15 for CECS and 13 for COCS) compared to 8 for BCS (7 for CECS and 8 for COCS) for patients who were not screened.

Fig 1 displays the performance of LSTM cancer screening predictions in terms of 10 repeated AUROCs for each type of screening. The average AUROC of 10 curves was 0.63 for BCS, 0.70 for CECS, and 0.61 for COCS.

Table 3 lists the numbers and proportions of patients in ideal, intermediate and poor categories for each submetric for the comparison between patients who received cancer screening and those who did not. We applied a Chi-squared test [29] to check if the frequencies (here percentages) between screening groups were significantly different from one other within each CVH submetric. As shown in Table 3, patients who received cancer screening had a higher

**Table 2. Characteristics [mean (SD) or n (%)] of the study population by receipt of cancer screening.**

| Cancer Screenings | Yes | | No | |
|---|---|---|---|---|
| BCS | n = 725 | | n = 8000 | |
| *Demographics* | | | | |
| Age, mean (std) year | 56 (8) | | 55 (10) | |
| White race, n (%) | 386 (53.3) | | 3875 (48.4)* | |
| Non-White race, n (%) | 136 (18.7) | | 1682 (21.0) | |
| Unknown race, n (%) | 203 (28.0) | | 2443 (30.5) | |
| *CVH factors mean (std), Avg # measures* | | *Avg #* | | *Avg #* |
| A1C | 7.1 (1.6) | 4.3 | 114.3 (37.5) | 3.4 |
| LDL (mg/dL) | 115.7 (37.1) | 3.0 | 32.9 (8.6) | 2.5 |
| BMI (kg/m$^2$) | 32.7 (7.3) | 9.8 | 127.4 (19.2) | 6.7 |
| Systolic blood pressure (SBP, mmHg) | 126.1 (17.8) | 11.2 | 77.4 (11.2) | 8.1 |
| Diastolic blood pressure (DBP, mmHg) | 77.1 (10.8) | 11.2 | 114.3 (37.5) | 8.1 |
| Current smoking n (%) | 184 (25.4) | 7.7 | 1802 (22.5) | 5.4 |
| CECS | n = 565 | | n = 3548 | |
| *Demographics* | | | | |
| Age, mean (std) year | 50 (7) | | 50 (8) | |
| White race, n (%) | 258 (45.6) | | 1550 (43.7) | |
| Non-White race, n (%) | 144 (25.4) | | 813 (22.9) | |
| Unknown race, n (%) | 163 (28.9) | | 1185 (33.4) | |
| *CVH factors mean (std), Avg # measures* | | *Avg #* | | *Avg #* |
| A1C | 7.2 (1.9) | 3.6 | 7.1 (1.9) | 2.9 |
| LDL (mg/dL) | 116.6 (36.0) | 3.1 | 115.3(36.6) | 2.2 |
| BMI (kg/m$^2$) | 30.6 (7.7) | 13.8 | 32.4 (8.6) | 6.3 |
| Systolic blood pressure (SBP, mmHg) | 124.3 (17.7) | 14.7 | 123.2 (18.6) | 7.3 |
| Diastolic blood pressure (DBP, mmHg) | 77.5 (10.6) | 14.7 | 76.6 (11.4) | 7.3 |
| Current smoking n (%) | 206 (36.5) | 10.1 | 838 (23.6) | 5.0 |
| COCS | n = 240 | | n = 3000 | |
| *Demographics* | | | | |
| Age, mean (std) year | 60 (7) | | 61 (8) | |
| White race, n (%) | 116 (48.4) | | 1698 (56.6) | |
| Non-White race, n (%) | 45 (18.9) | | 456 (15.2) | |
| Unknown race, n (%) | 79 (32.8) | | 846 (28.2) | |
| *CVH factors mean (std), Avg # measures* | | *Avg #* | | *Avg #* |
| A1C | 7.0 (1.4) | 4.8 | 7.0 (1.7) | 3.5 |
| LDL (mg/dL) | 105.4 (32.5) | 3.6 | 106.1 (38.1) | 2.9 |
| BMI (kg/m$^2$) | 31.6 (7.3) | 9.8 | 31.7 (7.7) | 6.4 |
| Systolic blood pressure (SBP, mmHg) | 129.8 (18.5) | 12.6 | 129.4 (19.4) | 8.0 |
| Diastolic blood pressure (DBP, mmHg) | 76.8 (11.3) | 12.6 | 76.9 (11.4) | 8.0 |
| Current smoking n (%) | 71 (29.6) | 8.4 | 713 (23.8) | 5.4 |

* The percentages may not add up to 100% due to rounding.

prevalence of poor A1C (62% for BCS, 58% for CECS and 72% for COCS) compared to patients who did not receive screening (53% for BCS, 53% for CECS and 51% for COCS).

Fig 2 shows changes in CVH submetrics within the same patient screening groups. Fig 2 (A)–2(C) show the changes in CVH submetrics for the patients who were screened, while Fig 2(E) and 2(F) show the changes in CVH for patients who were not screened.

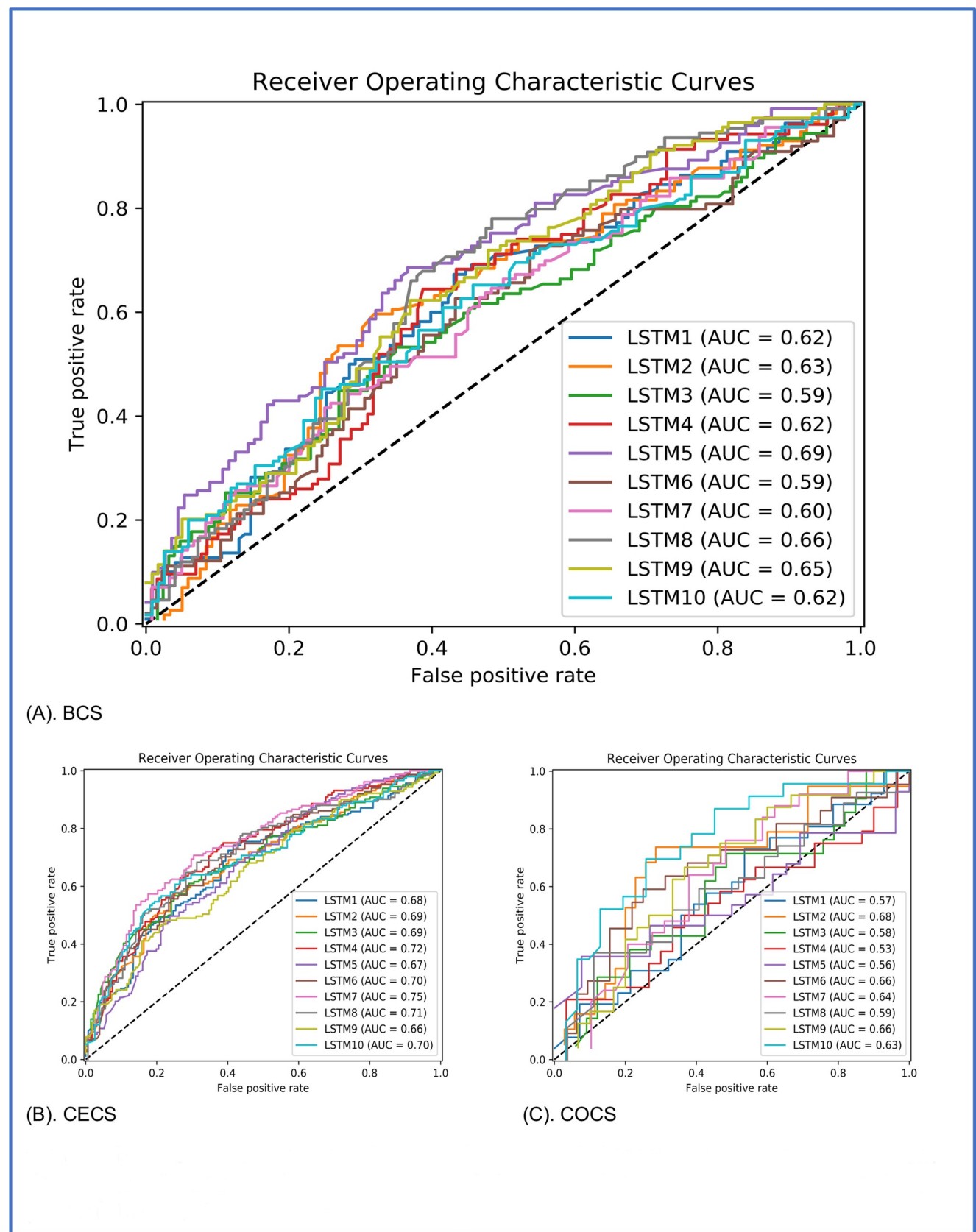

(A). BCS

(B). CECS

(C). COCS

**Fig 1.** The area under the curve (AUC) are shown for LSTM cancer screening predictions from time-series CVH factors which were repeated 10 times with different comparison patients for BCS (A), CECS (B) and COCS (C).

From the first column of Fig 2, we can see that the prevalence of "poor" submetrics decreased after cancer screenings. For example, all five submetrics improved after BCS (Fig 2 (A)), while BP and A1C improved after CECS (Fig 2(B)), and BP, A1C, and smoking improved after COCS (Fig 2(C)). Notably, for the prevalence of poor A1C decreased for all patients who received cancer screenings: 7% in BCS, 14% in CECS, and 17% in COCS. On the other hand, from the second column of Fig 2, we can see that the prevalence of "poor" A1C increased for all comparison patients.

## Discussion

In this study, we demonstrated associations between time-series CVH risk factor measures and receipt of three types of cancer screenings, i.e., breast, cervical, and colon cancer screenings, by using a nationally representative dataset–TGA data. The TGA data enabled us to examine multiple sites, CVH submetrics, and types of cancer screenings using advanced deep learning models. An advantage of our study was that all 5 CVH submetrics were investigated simultaneously for an association with 3 different cancer screenings on a unique nationally representative dataset of patients, i.e., the large TGA data set, which contains longitudinal

**Table 3. Comparison CVH factors between patients with cancer screening or without [n (%)].**

| Patients with BCS (n = 725) | BMI | BP | A1C | LDL | Smoking |
|---|---|---|---|---|---|
| Chi-squared p-value | 0.70 | 0.87 | 0.20 | 0.56 | 0.89 |
| Ideal | 185 (13.8) | 797 (28.5) | 55 (11.9) | 505 (68.2) | 1747 (72.8) |
| Intermediate | 315 (23.5) | 1350 (48.2) | 120 (25.9) | 130 (17.6) | 3 (0.1) |
| Poor | 838 (62.6) | 653 (23.3) | 288 (62.2) | 105 (14.2) | 650 (27.1) |
| Patients without BCS (n = 8000) | | | | | |
| Ideal | 5308 (16.8) | 17104 (28.8) | 1168 (14.4) | 8963 (69.4) | 29449 (74.5) |
| Intermediate | 7517 (23.8) | 27254 (45.9) | 2632 (32.4) | 2527 (19.6) | 96 (0.2) |
| Poor | 18761(59.4) | 15050 (25.3) | 4324 (53.2) | 1426 (11.0) | 9968 (25.2) |
| Patients with CECS (n = 565) | BMI | BP | A1C | LDL | Smoking |
| Chi-squared p-value | 0.60 | 0.65 | 0.14 | 0.72 | 0.97 |
| Ideal | 325 (21.6) | 1117 (36.3) | 49 (15.5) | 458 (68.8) | 1532 (66.4) |
| Intermediate | 481 (32.0) | 1323 (43.0) | 83 (26.2) | 126 (18.9) | 0 (0.0) |
| Poor | 699 (46.4) | 639 (20.8) | 185 (58.4) | 82 (12.3) | 775 (33.6) |
| Patients without CECS (n = 3548) | | | | | |
| Ideal | 2009 (17.4) | 8269 (32.6) | 512 (18.1) | 3224 (67.0) | 11984 (72.9) |
| Intermediate | 2773 (24.0) | 11243 (44.3) | 807 (28.6) | 994 (20.6) | 40 (0.2) |
| Poor | 6789 (58.7) | 5884 (23.2) | 1502 (53.2) | 596 (12.6) | 4422 (26.9) |
| Patients with COCS (n = 240) | BMI | BP | A1C | LDL | Smoking |
| Chi-squared p-value | 0.99 | 0.54 | 0.00 | 0.37 | 0.09 |
| Ideal | 102 (18.2) | 252 (23.2) | 22 (9.6) | 254 (81.4) | 744 (82.9) |
| Intermediate | 153 (27.3) | 478 (44.1) | 43 (18.7) | 42 (13.5) | 2 (0.2) |
| Poor | 306 (54.5) | 354 (32.7) | 165 (71.7) | 16 (5.1) | 152 (16.9) |
| Patients without COCS (n = 3000) | | | | | |
| Ideal | 2098 (17.7) | 5553 (25.3) | 538 (15.1) | 4533 (76.5) | 10948 (73.2) |
| Intermediate | 3235 (27.3) | 10328 (47.0) | 1198 (33.6) | 869 (14.7) | 34 (0.2) |
| Poor | 6516 (55.0) | 6101 (27.8) | 1828 (51.3) | 522 (8.8) | 3972 (26.6) |

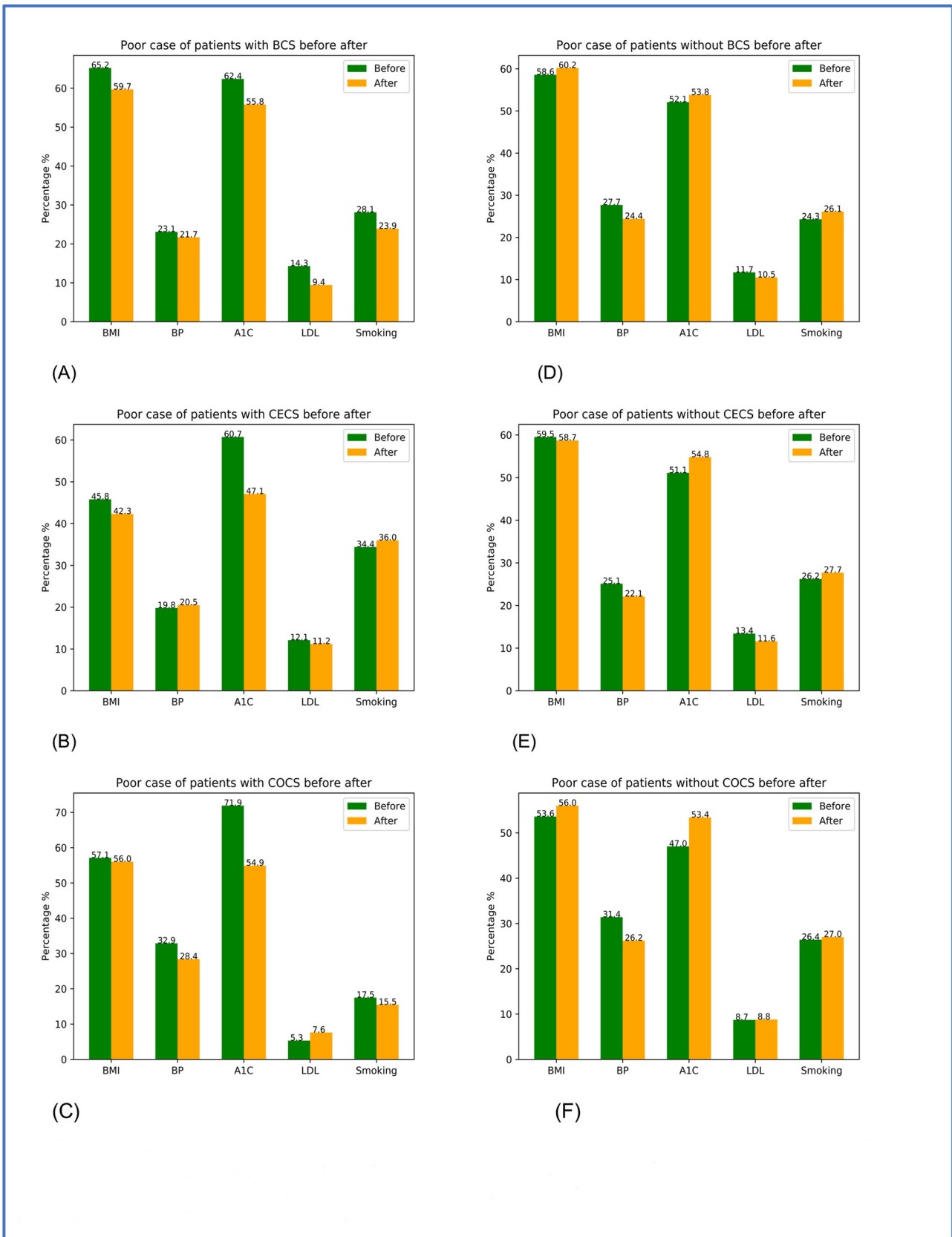

**Fig 2.** The plots of percentages for poor CVH factors for the same patients before and after time points of cancer screening for patients with screenings (A)–(C) and before and after middle time points for patients without cancer screenings (D)–(F). The first row is for BCS, second row is for CECS and the third is for COCS.

CVH measurements and cancer screening patterns from more than 70 different clinics in the US.

The comparison of different CVH measure distributions between patients who received cancer screenings and those who did not showed that patients with poorer CVH (especially poor A1C) were more likely to receive cancer screenings. Specifically, patients with poorer A1C were more likely to receive cancer screenings. Some recent studies have showed that individuals with diabetes had 30% higher incidence of certain cancers and also were more likely to be diagnosed with advanced-stage tumors [30–33]. Thus, providers might be more likely to recommend patients with diabetes to uptake cancer screenings for early prevention of developing cancers, which may lead to more individuals with diabetes to participate in cancer screenings.

Moreover, we investigated the effects of cancer screenings on the changes of CVH measures of the patients to better understand if the screenings had potential associations with the improvement of CVH measures. Our results indicated that patients who received cancer screenings appeared to have better control of CVH factors, especially A1C, than patients who did not receive cancer screenings. Specifically, A1C levels were improved after patients received any type of screening, while A1C levels worsened among patients who did not receive cancer screening. A similar trend could be observed for BMI: it became better after patients received any type of screening, while BMI became worse among patients without BCS or COCS. Levels of BP were improved after patients received BCS or COCS screenings and worsened among patients without BCS or COCS. Poor levels of LDL decreased among patients after receipt of BCS and among those without BCS. However, LDL improvements were much greater among patients after receipt of BCS (34% decrease in LDL) than those without BCS (10% decrease in LDL). After receipt of BCS and COCS, current smoking declined compared to the increase observed among those without the screenings.

In summary, our analyses showed that patients with poor CVH measures were more likely to receive cancer screenings. Patients with receipt of cancer screenings appeared to have improved CVH measures after the screening as compared to before. One possible reason for this was that patients might receive more attention and through care from providers to detect and manage CVH by virtue of reviewing cancer screening and other risk factor data. At the population level, better CVH is associated with a lower risk of cardiovascular disease (CVD) and cancers [34,35]. Thus, cancer screenings may indirectly decrease burden and cost on the health system (e.g., CVD and cancers) by improving patient CVH health.

## Limitations

There were some limitations in our analyses. We used values of AUROC to evaluate associations between time-series CVH measurements and receipt of cancer screenings. Higher AUROC values indicated stronger associations between predictors and the binary outcomes [36]. However, our observed AUROC values were relatively low and thus have limited clinical utility at this time. Cancer screenings are potentially affected by CVH and other factors. We acknowledge that we had relatively few patients with receipt of cancer screening. Specifically, there were relatively few patients who received cancer screenings compared to patients who did not within the same age and gender groups. This limitation likely affected the accuracy of

our prediction models. The prediction accuracy of our models could be improved if more patients in our data set had received cancer screening.

## Conclusions

We demonstrated that deep learning LSTM models can effectively predict the associations between time-series CVH measures and receipt of cancer screening. Poor CVH, especially poor A1C, may prompt providers to recommend cancer screening for their patients. And patients who received cancer screening may also receive better care for and/or have improved self-management of CVH, especially A1C. Overall, these findings suggest that unhealthier patients are screened for cancers, and that cancer screening may also prompt favorable changes in CVH.

## Author Contributions

**Conceptualization:** Randi E. Foraker.

**Formal analysis:** Aixia Guo.

**Supervision:** Randi E. Foraker.

**Writing – original draft:** Aixia Guo.

**Writing – review & editing:** Bettina F. Drake, Yosef M. Khan, James R. Langabeer II, Randi E. Foraker.

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
