## [Decision Letter · Decision Letter 0]

5 Jun 2020

PONE-D-20-10777

Time-series cardiovascular risk factors and receipt of screening for breast, cervical, and colon cancer: The Guideline Advantage

PLOS ONE

Dear Dr. Foraker,

Thank you for submitting your manuscript to PLOS ONE. After careful consideration, we feel that it has merit but does not fully meet PLOS ONE’s publication criteria as it currently stands. Therefore, we invite you to submit a revised version of the manuscript that addresses the points raised during the review process.

We look forward to receiving your revised manuscript.

Kind regards,

Antonio Palazón-Bru, PhD

Academic Editor

PLOS ONE

Journal Requirements:

2. In ethics statement in the manuscript and in the online submission form, please provide additional information about the patient records used in your retrospective study. Specifically, please ensure that you have discussed whether all data were fully anonymized before you accessed them and/or whether the IRB or ethics committee waived the requirement for informed consent. If patients provided informed written consent to have data from their medical records used in research, please include this information.

4. Your ethics statement must appear in the Methods section of your manuscript. If your ethics statement is written in any section besides the Methods, please move it to the Methods section and delete it from any other section. Please also ensure that your ethics statement is included in your manuscript, as the ethics section of your online submission will not be published alongside your manuscript.

5. We note you have included a table to which you do not refer in the text of your manuscript. Please ensure that you refer to Table 2 in your text; if accepted, production will need this reference to link the reader to the Table.

6. Please include a caption for figure 3.

Reviewers' comments:

Reviewer's Responses to Questions

**Comments to the Author**

1. Is the manuscript technically sound, and do the data support the conclusions?

Reviewer #1: Yes

2. Has the statistical analysis been performed appropriately and rigorously? 

Reviewer #1: Yes

3. Have the authors made all data underlying the findings in their manuscript fully available?

Reviewer #1: No

4. Is the manuscript presented in an intelligible fashion and written in standard English?

Reviewer #1: Yes

5. Review Comments to the Author

Reviewer #1: The authors investigate time-series cardiovascular risk factors and the association with cancer screening.

Major

- Average AUC values are not listed in the result section. Please do so.

- Discussion, please do not repeat your introduction in the first paragraph but list your main findings instead.

- The discussion is short and interpretation of results is limited. Please expand.

- AUC of models is quite limited, the authors should address this.

- It should be recommended to post the deep learning protocol at repositories like github, for other users to test.

Minor

- Introduction is on the long side please reduce to a single page.

- Avoid speaking language like “Fortunately”

6. PLOS authors have the option to publish the peer review history of their article (what does this mean?). If published, this will include your full peer review and any attached files.

Reviewer #1: No

---

## [Author Response · Author response to Decision Letter 0]

6 Jul 2020

Response to reviewer comments

We would like to thank reviewers for the helpful comments. Please find our responses to the reviews below in bold. We hope you find the revised manuscript suitable for publication.

Journal Requirements:

Thank you for the instructions, we have modified the manuscript according to the instructions.

2. In ethics statement in the manuscript and in the online submission form, please provide additional information about the patient records used in your retrospective study. Specifically, please ensure that you have discussed whether all data were fully anonymized before you accessed them and/or whether the IRB or ethics committee waived the requirement for informed consent. If patients provided informed written consent to have data from their medical records used in research, please include this information.

Thank you. We have added the details “All the data were fully anonymized before we accessed them.”

Thank you. In our Data Availability statement, we stated that we will be able to share the Python code related to the analyses at the time of acceptance of the manuscript. As the data owned by a third party, we do not have permission to share the data. Requesting access to The Guideline Advantage (TGA) data must be requested from the American Heart Association.

4. Your ethics statement must appear in the Methods section of your manuscript. If your ethics statement is written in any section besides the Methods, please move it to the Methods section and delete it from any other section. Please also ensure that your ethics statement is included in your manuscript, as the ethics section of your online submission will not be published alongside your manuscript.

Thank you. We have moved the ethics statement to the beginning of the Methods section.

5. We note you have included a table to which you do not refer in the text of your manuscript. Please ensure that you refer to Table 2 in your text; if accepted, production will need this reference to link the reader to the Table.

Thank you for the great comment. We have included the Table 2 in the text.

6. Please include a caption for figure 3.

Thank you. We have corrected figure 3 to figure 2 which included a caption.

Reviewers' comments:

Reviewer's Responses to Questions

Comments to the Author

1. Is the manuscript technically sound, and do the data support the conclusions?

Reviewer #1: Yes

2. Has the statistical analysis been performed appropriately and rigorously? 

Reviewer #1: Yes

3. Have the authors made all data underlying the findings in their manuscript fully available?

Reviewer #1: No 

Thank you for the point. Please find the data availability statement in our manuscript as follows. 

“Data availability statement

The data are owned by a third party and the authors do not have permission to share the data. Requesting access to The Guideline Advantage (TGA) data must be requested from the American Heart Association. The study team will be able to share the code related to the analyses at the time of acceptance of the manuscript.”

4. Is the manuscript presented in an intelligible fashion and written in standard English?

Reviewer #1: Yes

5. Review Comments to the Author

Reviewer #1: The authors investigate time-series cardiovascular risk factors and the association with cancer screening.

Major

- Average AUC values are not listed in the result section. Please do so.

Thank you for the great point. We have added the average AUC values in the result section.

- Discussion, please do not repeat your introduction in the first paragraph but list your main findings instead.

Thank you for the great point. We have modified the first paragraph by listing our main findings in discussion section.

- The discussion is short and interpretation of results is limited. Please expand.

Thank you for the great point. We have expanded the discussion section and interpreted more of the results.

- AUC of models is quite limited, the authors should address this.

Thank you, this is a great point. We have addressed this limit in the limitation section.

- It should be recommended to post the deep learning protocol at repositories like github, for other users to test.

Thank you for the great suggestion. We have created a github repository and have posted the related Python code here: https://github.com/aixiaguo/Python-Code

Minor

- Introduction is on the long side please reduce to a single page.

Thank you, we reduced the introduction to one page. 

- Avoid speaking language like “Fortunately”

Thank you, we have deleted the speaking language similar to “Fortunately”.

---

## [Decision Letter · Decision Letter 1]

10 Jul 2020

PONE-D-20-10777R1

Time-series cardiovascular risk factors and receipt of screening for breast, cervical, and colon cancer: The Guideline Advantage

PLOS ONE

Dear Dr. Foraker,

Thank you for submitting your manuscript to PLOS ONE. After careful consideration, we feel that it has merit but does not fully meet PLOS ONE’s publication criteria as it currently stands. Therefore, we invite you to submit a revised version of the manuscript that addresses the points raised during the review process.

We look forward to receiving your revised manuscript.

Kind regards,

Antonio Palazón-Bru, PhD

Academic Editor

PLOS ONE

Reviewers' comments:

Reviewer's Responses to Questions

**Comments to the Author**

1. If the authors have adequately addressed your comments raised in a previous round of review and you feel that this manuscript is now acceptable for publication, you may indicate that here to bypass the “Comments to the Author” section, enter your conflict of interest statement in the “Confidential to Editor” section, and submit your "Accept" recommendation.

Reviewer #1: All comments have been addressed

2. Is the manuscript technically sound, and do the data support the conclusions?

Reviewer #1: Yes

3. Has the statistical analysis been performed appropriately and rigorously? 

Reviewer #1: Yes

4. Have the authors made all data underlying the findings in their manuscript fully available?

Reviewer #1: No

5. Is the manuscript presented in an intelligible fashion and written in standard English?

Reviewer #1: Yes

6. Review Comments to the Author

Reviewer #1: The authors addressed most comments to satisfaction. One minor comment:

Please start discussion listing the most important finding: “In this

study, we demonstrated associations between time-series” , “Regular cancer … after cancer screenings” can be deleted.

7. PLOS authors have the option to publish the peer review history of their article (what does this mean?). If published, this will include your full peer review and any attached files.

Reviewer #1: No

---

## [Author Response · Author response to Decision Letter 1]

10 Jul 2020

Please start discussion listing the most important finding: “In this study, we demonstrated associations between time-series”, “Regular cancer … after cancer screenings” can be deleted.

Thank you. We really appreciate your great comments. 

We have deleted “Regular cancer … after cancer screenings”.

---

## [Decision Letter · Decision Letter 2]

15 Jul 2020

Time-series cardiovascular risk factors and receipt of screening for breast, cervical, and colon cancer: The Guideline Advantage

PONE-D-20-10777R2

Dear Dr. Foraker,

We’re pleased to inform you that your manuscript has been judged scientifically suitable for publication and will be formally accepted for publication once it meets all outstanding technical requirements.

Kind regards,

Antonio Palazón-Bru, PhD

Academic Editor

PLOS ONE

Additional Editor Comments (optional):

Reviewers' comments:

Reviewer's Responses to Questions

**Comments to the Author**

1. If the authors have adequately addressed your comments raised in a previous round of review and you feel that this manuscript is now acceptable for publication, you may indicate that here to bypass the “Comments to the Author” section, enter your conflict of interest statement in the “Confidential to Editor” section, and submit your "Accept" recommendation.

Reviewer #1: All comments have been addressed

2. Is the manuscript technically sound, and do the data support the conclusions?

Reviewer #1: Yes

3. Has the statistical analysis been performed appropriately and rigorously? 

Reviewer #1: Yes

4. Have the authors made all data underlying the findings in their manuscript fully available?

Reviewer #1: No

5. Is the manuscript presented in an intelligible fashion and written in standard English?

Reviewer #1: Yes

6. Review Comments to the Author

Reviewer #1: Th authors greatly improved the manuscript. AL comments have been addressed to satisfaction. No further remakes,

7. PLOS authors have the option to publish the peer review history of their article (what does this mean?). If published, this will include your full peer review and any attached files.

Reviewer #1: No

---

## [Editor Report · Acceptance letter]

3 Aug 2020

PONE-D-20-10777R2 

Time-series cardiovascular risk factors and receipt of screening for breast, cervical, and colon cancer: The Guideline Advantage 

Dear Dr. Foraker:

I'm pleased to inform you that your manuscript has been deemed suitable for publication in PLOS ONE. Congratulations! Your manuscript is now with our production department. 

Kind regards, 

on behalf of

Dr. Antonio Palazón-Bru 

Academic Editor

PLOS ONE